# On The Effectiveness-Fluency Trade-Off In LLM Conditioning: A Systematic Study

## Abstract

Controlling the output of Large Language Models (LLMs) is a central challenge for their safe and reliable deployment, yet a clear understanding of the trade-offs involved remains elusive. Current approaches to conditioning generation, spanning from expensive fine-tuning to lightweight activation steering or basic prompting, are often evaluated with a narrow focus on their effectiveness at injecting or removing a target concept, neglecting critical side effects on the quality of the generation. This paper presents a systematic investigation of these methods in both injection and removal scenarios, introducing a comprehensive evaluation framework to assess generation quality and move beyond unreliable measures like perplexity. Our analysis reveals that the latter is a fundamentally brittle proxy for fluency, often rewarding repetitive text while penalizing well-formed outputs. Using more robust metrics, we find that there is no "free lunch" in conditioning: lightweight methods frequently achieve conditioning at a steep cost to expressiveness. Furthermore, we identify a critical yet previously overlooked interaction with the training paradigm: activation steering methods are far less effective on instruction-tuned models than on their base counterparts. While supervised fine-tuning emerges as the most robust method, it also exhibits significant side effects, such as collateral learning of possibly undesired linguistic characteristics of the training set. Finally, simple prompting might be an alternative to more sophisticated conditioning methods for basic concept injection, but it fails to scale to tasks requiring more thorough output control, such as concept removal. Collectively, our findings challenge common assumptions in the field, providing a more realistic characterization of the conditioning landscape and a simple but principled methodology for future evaluation.

## 1 Introduction

Large Language Models (LLMs) trained on vast amounts of text have led to a revolution in language processing (Raffel et al., 2020; Brown et al., 2020). Their widespread use has highlighted the need to "adjust", or *condition* LLM generations to either inject or remove (un)desired *concepts*, such as persons, or topics (*e.g.,* Winston Churchill, dogs, or trees), but also more abstract properties such as generation language or style (*e.g.,* French, formality, or toxicity).

Approaches for conditioning LLM generations vary along several complexity axes, including computational cost, amount of data and labels, and how accessible these solutions are to a user. Such methods range from (i) highly complex supervised fine-tuning (Ouyang et al., 2022; Almasi & Schiønning, 2023), to (ii) medium-complexity activation steering (Li et al., 2024; Rodriguez et al., 2025a; Rimsky et al., 2023), to (iii) simple prompt engineering (Marvin et al., 2023).

While many studies focus primarily on the conditioning effectiveness (*i.e.,* successfully including or excluding a concept), they often neglect the impact on generation quality, such as fluency and grammatical correctness. The analysis of these side effects is frequently limited to perplexity scores (Rodriguez et al., 2025a; Li et al., 2024) or qualitative analysis of a few examples (Rimsky et al., 2023).

We take here a more holistic approach, by investigating conditioning methods across all three categories (fine-tuning, steering and prompting) with an evaluation strategy that extends beyond the aforementioned standard metrics. We incorporate a comprehensive set of linguistic evaluations to analyze the content and style of the generated outputs as well as a more costly, LLM-judged fluency

assessment. This comprehensive analysis provides a clearer understanding of the overall utility and trade-offs of different conditioning methods.

In short, our main contributions are as follows:

- **A comprehensive analysis of conditioning methods** (fine-tuning, activation steering, and prompting) using a suite of metrics that goes beyond perplexity to evaluate generation quality and fluency.

- **A demonstration that perplexity is an unreliable proxy for generation quality**, as it can be skewed by repetition or sentence length. Simpler metrics, such as the type-token ratio, correlate more closely with human judgments of fluency.

- **The discovery of a critical interaction between conditioning and model type**, revealing that the effectiveness of activation steering is strongly reduced on instruction-tuned models.

- **An example of how the nature of the conditioning task differentially affects conditioning methods**, with supervised fine-tuning and direct prompting out-performing activation steering methods on concept injection, but lagging behind in a concept removal task (toxicity mitigation).

## 2    RELATED WORK

**Conditioning approaches**    As LLMs become increasingly powerful and widely deployed, a significant body of work has emerged to condition their outputs toward desired specifications. LLM conditioning methods range in complexity. The most straightforward technique is prompting, where instructions are provided directly in the input. However, reliance on prompting alone is often insufficient, as it provides limited fine-grained control and can be an unreliable strategy for achieving precise behaviors. Full fine-tuning strategies (Wei et al., 2022) or reinforcement learning from human feedback (Ouyang et al., 2022; Bai et al., 2022) have been proposed to adapt model behavior, with the issue of being data and compute intensive (Casper et al., 2023). More lightweight fine-tuning solutions (Hu et al., 2022) have become very popular. Activation steering has emerged as an even lighter and adaptable strategy for conditioning. Earlier methods proposed to add a steering vector on intermediate activations. Such vector can be estimated as a difference of means (Rimsky et al., 2023), tangent to a binary classifier hyperplane (Li et al., 2024), or as the difference between the representations of two preprompts computed during generation (Zou et al., 2023), among others. Other methods focus on intervening on expert neurons (Suau et al., 2022), which has shown to be an effective strategy (Fedzechkina et al., 2025). For example, Suau et al. (2024) mitigate toxicity by dampening the activations of expert neurons according to their expertise. More recent methods propose to intervene on activations using affine maps optimized offline using an optimal transport loss (Rodriguez et al., 2025a) or even optimized jointly for all layers in the LLM (Rodriguez et al., 2025b). Alternatively, Wu et al. (2024) proposed a low-rank linear map optimized on the language modeling objective.

While current methods for conditioning LLMs are evaluated for their effectiveness and impact on general benchmarks, their fine-grained effects on language generation remain largely unexplored. Existing evaluation frameworks typically rely on classifiers or LLM judges to measure concept adherence, supplemented by perplexity or Massive Multitask Language Understanding (MMLU) scores (Hendrycks et al., 2021) . Perplexity and MMLU are often used as "guardrail" metrics to check for general performance degradation. As we will show below, perplexity is not a reliable measure of fluency, as it can be low for trivially predictable text (e.g., repetitions) and high for perfectly fluent but stylistically complex text. Concurrently, MMLU's focus on factual recognition (across 57 subjects in a multiple-choice format) is orthogonal to typical conditioning goals like style or persona, and its multiple-choice format fails to assess changes in generative behavior. While these metrics can detect catastrophic failures, alone they are limited when it comes to assessing the nuanced linguistic success or subtle negative side effects of the conditioning itself, leaving a critical gap in evaluation.

Wu et al. (2025) present the closest work to ours, focusing on evaluation of conditioning methods in a concept injection task. They evaluate various methods on the geometric average of 3 LLM-as-judge measures: concept presence, fluency and instruction-following. Like us, they find that simple prompting and supervised fine-tuning outperform some state-of-the-art steering methods, such as

Sparse Autoencoders (SAEs). With respect to their evaluation, besides focusing on newer state-of-the-art methods, we bring new insights on the effect of control on LLMs by clearly distinguishing the evaluation of successful conditioning (concept presence/relevance) from that of fluency, we report the discovery that instruction-tuning makes models remarkably resistant to activation steering, and we show that, while prompting and supervised fine-tuning do indeed outperform activation steering at concept *injection*, they fail at the task of concept *removal*.

**LLM fluency evaluation**   Outside the specific domain of evaluating control methods, the problem of perplexity/likelihood as an unreliable measure of generation quality in language model evaluation has been observed multiple times. Concurrent early work by Holtzman et al. (2020) and Welleck et al. (2020) found that likelihood-maximization training objectives led to the generation of highly predictable, repetitive and generic text. To quantify the problem, they used simple corpus-based statistical measures similar to the ones we use here. Pillutla et al. (2021) proposed the MAUVE score to assess the quality of open-ended text generation by comparing an LLM distribution to that of human-generated text. We prefer to focus on methods that do not require a reference human distribution, as in many control scenarios it might be difficult to come up with such a reference (*e.g.*, if we want the model to target a specific concept, it would be complicated and expensive to produce human generated-text for all legitimate contexts and styles in which the LLM could generate text on the target concept). Similarly, while extended analyses such as those of Meister & Cotterell (2021) and He et al. (2023) are very informative about the nature of LLM generated text, our focus is to identify simple measures that can pinpoint issues with text naturalness in the context of evaluating conditioning methods.

# 3   METHODS

In Section 3.1 we describe the conditioning methods considered, followed by a discussion on the data used to train such methods for different tasks (Section 3.2). Following state-of-the-art research on conditioning (Li et al., 2024; Rodriguez et al., 2025a; Zou et al., 2023), we focus on the tasks of (i) *concept injection*, where the LLM is conditioned to produce continuations centered on a concept of interest; and (ii) *concept removal*, where the objective is, instead, to produce continuations that do not contain the concept of interest (here, toxicity). To assess the global effect of each conditioning method, we propose a set of observables (Section 3.3), inspired by corpus linguistics and natural language generation evaluation, that capture complementary aspects of the generated text.

## 3.1   CONDITIONING METHODS

We briefly explain the different conditioning methods for controlling LLMs that will be compared through our pool of observables.

**Prompting**   For the concept injection task, we create 10 different prompting templates along the lines of: *"You are a chatbot that specializes in <concept>."* (refer to Appendix A.2 for the whole set). For toxicity mitigation, we resort to the prompts suggested by Suau et al. (2024), known to reduce toxic language in continuations. In both cases, such prompts are pre-pended to the generic prompts used to kickstart generation, which are described in Section 4.1.

**Supervised fine-tuning (SFT)**   We use LoRA adapters (Hu et al., 2022) as supervised fine-tuning technique. For that, we train LoRAs for each task on the attention modules of each LLM. LoRA training is performed by minimizing the language modeling loss (logit cross-entropy) over the target sentences of each concept. We train our LoRAs using AdamW (Loshchilov & Hutter, 2019) for 30 steps with learning_rate $= 10^{-4}$, LoRA rank $r = 2$, and dropout $= 0.05$ (see Appendix D for a brief discussion on the choice of hyperparameters).

We moreover experiment with three *activation steering* methods, namely:

**ITI (Li et al., 2024)**   ITI adds a bias to intermediate representations (activations). Such bias is learned as the tangent vector to a binary classifier that separates source activations (from text with undesired presence or absence of concept) and target activations (from text with desired presence or absence of concept) at each model layer. ITI uses last token pooling to train the classifier and intervenes only on the last tokens in the sequence. Following Li et al. (2024), ITI is used only on the attention output layers.

**CAA (Rimsky et al., 2023)** CAA subtracts last token source and target activations for multiple contrast pairs and averages them into a single steering vector. This vector is then added to all new generated tokens. We apply CAA on all attention output layers.

**Linear AcT (Rodriguez et al., 2025a)** In this work, the authors propose to minimize an optimal transport loss to bring the source activation distribution close to the target activation distribution via a linear map. Linear AcT uses mean pooling for training, and intervenes on all tokens at generation time. Following Rodriguez et al. (2025a), Linear AcT is applied on all LayerNorm layers.

In order to train the activation steering methods, following Rimsky et al. (2023); Li et al. (2024); Suau et al. (2024); Wu et al. (2024); Rodriguez et al. (2025a;b), we use small sets of source sentences (that do not contain the desired properties, *i.e.,* no concept / toxic) and target sentences (that do, *i.e.,* concept / non-toxic).

## 3.2    Training datasets

### 3.2.1    Concept Injection

To test the impact of conditioning methods for concept injection, we require a dataset of concepts that is representative enough. We follow the procedure described by Fedzechkina et al. (2025) and we generate a dataset of sentences that describe 60 topics, grouped in a hierarchical fashion into 10 categories. For instance, the category *color* contains finer concepts (*i.e.,* members) such as *red, green, blue, purple* or *black*. The concepts are chosen to cover wide semantic domains, in order to test the models in different areas and characterize their overall, concept-independent average behavior. We generate 400 sentences about each concept using Mistral-7B-instruct (Jiang et al., 2023). This quantity sufficed for successful convergence in supervised fine-tuning, while not being so large as to unfairly favor the supervised approach. More details can be found in Appendix B.

For each topic, the SFT conditioning is obtained by directly training the models on the 400 sentences. In the case of ITI, CAA and Linear AcT, this set represents the target sentences, while the source sentences are randomly sampled from all the other concepts.

### 3.2.2    Concept Removal (Toxicity Mitigation)

To test the impact of conditioning methods for concept removal, we adopt the task of toxicity mitigation. For the training phase, we use subsets of the Real Toxicity Prompts (RTP) dataset (Gehman et al., 2020). In particular, as source sentences for the activation steering methods, we collect entries with prompts and continuations whose reported toxicity is larger than 0.875 (for a total of 356 sentences) and as target sentences we extract those entries with prompts and continuation toxicity below 0.0275 (collecting 379 sentences).

For supervised fine-tuning, we build a subset whose prompt toxicity is larger than 0.875 and continuation toxicity smaller than 0.05, amounting to 455 sentences. The rationale is that the model would learn to provide a sane continuation even if the prompt or beginning of the sentence is toxic. The reported thresholds were chosen to obtain a similar number of samples in all the experiments.

## 3.3    Pool of Observables

We first define our success metrics for each task, namely *Concept Similarity* for concept injection and *Toxicity* for toxicity mitigation. Then, we define our pool of observables, aiming at capturing diverse linguistic aspects of the generated text.

**Concept Similarity (↑ *concept injection task success*)** Since the objective of concept injection is to condition model generation towards a concept, we quantify how much the target concept is present in the generated continuations as a measure of task success. To this aim, we embed both the concept name (a specific word such as "carrot" or "organ") and the generated continuations through Sentence-BERT (SBERT) (Reimers & Gurevych, 2019) equipped with the `all-mpnet-base-v2` model, and we compute their cosine similarity. Note that, by using this embedding-based score, we are not only measuring the direct occurrence rate of the concept name in the continuation, but whether the overall topic of the latter is aligned with the concept. The larger the Concept Similarity, the more effective

conditioning is at injecting the target concept. See Appendix C.1 for examples and further details, including a human validation of the measure.

**Toxicity** ($\downarrow$ *toxicity mitigation task success*)  For the concept removal / toxicity mitigation task, following Suau et al. (2024), we assess whether the generation after a given prompt is toxic or not through a ROBERTA-based binary classifier.[1] The final score is given by the fraction of toxic continuations. We expect lower toxicity after conditioning.

**Fluency**  Following Wu et al. (2025), we use a properly prompted LLM-as-a-judge to measure whether the continuation generated by the model after the intervention is still in fluent English. The model adopted for this task is OLMo2-32B-it (Walsh et al., 2025). Each continuation is assigned an integer value between 0 and 2. The final score for a method is obtained by averaging these values across the generations of the method (see Appendix C.3 for details, including human validation).

**Perplexity**  Perplexity (Jelinek et al., 1977) is a standard measure of the fluency of a language model, and it is often used in conditioning studies (Suau et al., 2024; Rodriguez et al., 2025a) to ensure that outputs remain "probable" after the intervention. As we do not know how conditioning methods affect the model output probability distribution (this is part of what we are trying to assess), the perplexity is always measured using an external model, again from the OLMo family, namely OLMo2-13B (Walsh et al., 2025). Some examples are provided in Appendix C.2.

**Type-Token Ratio**  A simple measure of lexical variety widely used in corpus linguistics (Lüdeling & Kytö, 2008) is the ratio between the number of unique words and the total word count in a corpus, which in our case is given by the set of samples generated under conditioning. The higher this ratio is, the less repetitive the vocabulary used by the LLM.

**Generation Similarity**  The Type-Token Ratio captures direct word repetitions, but a broader measure of generation monotony should account for the tendency of a conditioned LLM to produce continuations that are semantically close, regardless of the specific words used. For this purpose, inspired by the recent use of embedding-based metrics in Natural Language Generation (Sai et al., 2022), we embed the continuations through SBERT and compute their average pairwise cosine similarity. The larger this value, the more similar the generated continuations and, consequently, the less expressive the conditioned model. See Appendix C.1 for more details and examples.

**POS KL**  The goal of concept injection/removal is typically to affect the meaning of generations, but the intervention could also affect the kind of *style* models are using, and in particular their choices of syntactic expression (*e.g.,* using a more descriptive, nominal style, or a more narrative one heavy on verbs). To assess how much the interventions alter the linguistic structures learned and used by the model, we compute the Kullback-Leibler (KL) divergence (Kullback & Leibler, 1951) between the distribution of Parts Of Speech (*i.e.,* word categories such as noun and verb, abbreviated as POS) in the continuations generated by the base model and the ones generated by a conditioned variant. The POS are obtained through the SpaCy tagger (Honnibal et al., 2020). The smaller the POS KL value, the less the syntactic structures are affected by conditioning (which should generally be desirable).

**Generation Length**  Finally, we record the number of tokens produced by the model, that can always insert an EOS token (marking the end of the generation) before it reaches the maximum 100-token limit we impose on its generations. Intuitively, we would like post-intervention continuations to have similar length to pre-intervention continuations.

## 4  RESULTS

In our experiments, we use the following 3 LLMs in both their base pre-trained versions as well as the instruction-tuned (IT) variants:[2] Qwen3-0.6B, Qwen3-8B (Yang et al., 2025), Smollm3-3B (Bakouch et al., 2025). We fix the following sampling parameters: $\text{temperature} = 1.0, \text{top\_p} = 0.6, \text{repetition\_penalty} = 1.0$.

---

[1] https://huggingface.co/s-nlp/roberta_toxicity_classifier

[2] See Appendix A.4 for the chat template implementation details.

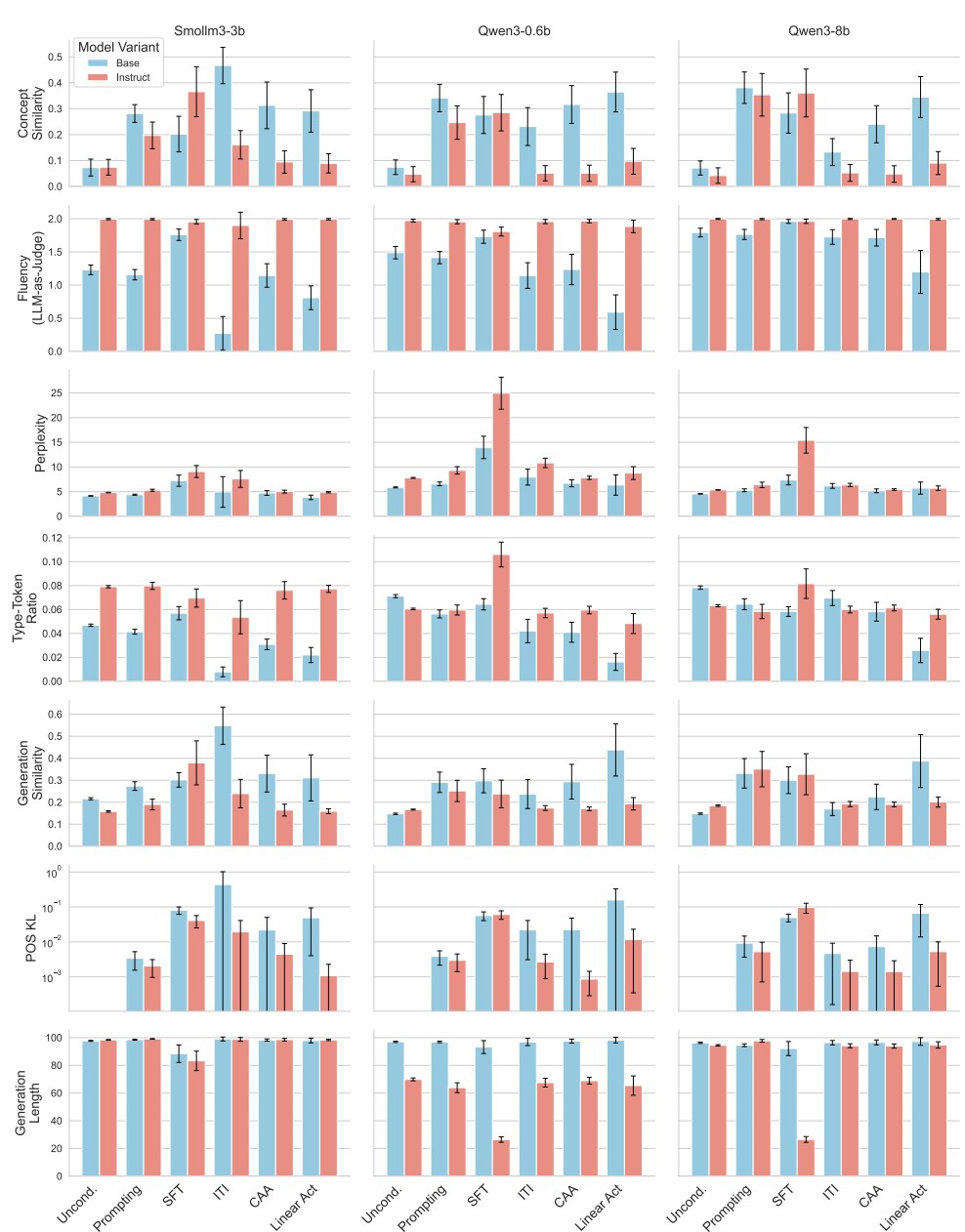

Figure 1: Our set of observables for the conditioning methods applied to the different models in the concept injection experiments. We report averages (and standard deviations) for measures obtained on each concept individually.

## 4.1 CONCEPT INJECTION

**Setup**    We create a set of 100 generic, diversified and concept-free prompts, along the line of: *Describe a place, Tell a story, Set a scene*.[3] For each concept+conditioning+model combination, these generic prompts are randomly sampled and used to generate 1000 continuations composed of up to 100 tokens. In pure prompting induction, conditioning is achieved by pre-pending randomly sampled concept-inducing sentences (described in Section 3.2) to these generic prompts. For supervised fine-tuning and steering, the generic prompts are simply passed as inputs to the conditioned model.

---

[3]The full set and other details are reported in Appendix A.1

The observables are computed separately for each concept and in Figure 1 we report the resulting averages (and standard deviations).

**Conditioning effectiveness in activation steering comes at a fluency cost**   Across the board, when steering methods achieve high Concept Similarity (first row), *i.e.,* they accomplish the task of producing concept-relevant continuations,[4] this comes at the cost of a significant loss in the Fluency score (second row). Linear AcT, in particular, achieves high Concept Similarity for all non-instructed models, but it also displays consistently low fluency. ITI has a strong peak in Concept Similarity for Smollm3-3B, but this is again accompanied by very low fluency.

**Instruction-tuning hinders steering**   The activation steering methods have higher fluency when applied to IT models, but their Concept Similarity scores are rarely above those of the base model, suggesting that steering has not really succeeded. Indeed, qualitatively, we find the continuations generated by activation-steered IT models to be virtually indistinguishable from those of the corresponding base model. By inspecting the outputs, the effect seems due to the fact that IT models are extremely good at following the generic prompt we feed them, at the cost of ignoring the push towards generating concept-relevant text. For example, if the prompt is "Imagine a world", an IT model will be much better than its uninstructed counterpart in terms of world building, but at the price of ignoring the imperative to talk about the target concept. This effect is also present, but to a smaller degree, for prompting, whereas it does not affect the ability of supervised fine-tuning (SFT) to produce concept-relevant text. Interestingly, though, SFT applied to IT models results in much shorter continuations (last row) than under any other form of conditioning.

**Perplexity is an unreliable measure of fluency**   Perplexity (third row) should be negatively correlated with the Fluency score, but actually, across all model+conditioning combinations, it is weakly *positively* correlated with it ($\rho = 0.17$). Indeed, perplexity is consistently remarkably high for SFT, despite this method also consistently achieving some of the highest Fluency scores. Qualitatively, we observe that the high Perplexity scores of SFT are not due to poor text generation, but rather to pathologies in what perplexity is capturing. For example, SFT has a much higher perplexity than the other methods for Qwen3-0.6B. Inspecting the generated continuations, we see that this is due to the fact that it tends to produce fanciful, stylistically complex text, which is harder to predict for the reference model. As a concrete case, when prompted to "tell a myth", with *carrot* as target concept, SFT-conditioned Qwen3-0.6B produces an elaborated, literary-sounding story that starts with: "A long, dark night saw a lone carrot thrive, its deep orange skin hidden beneath a secret tapestry of nutrient." Moreover, SFT perplexity for IT models tend to be high simply because, on this class of models, as we noted above, SFT-conditioning produces rather short continuations. As perplexity tends to be higher at the beginning of a sentence, shorter sentences tend to have higher perplexity (Jurafsky & Martin, 2023). On the other hand, activation steering methods often have low perplexity out of producing very repetitive, and hence highly predictable text. For example, Linear AcT has remarkably low perplexity when applied to Smollm3-3B, but a look at the generations reveals that this is due to degenerate text such as "If you are a carrot, then you are a carrot. If you are a carrot, then you are a carrot...". More examples in Appendix C.2.

**The Type-Token Ratio is a good cue of fluency**   When looking at score correlations across model+conditioning combinations, we find that the very simple Type-Token Ratio score (fourth row) is by far the observable that is most correlated with the Fluency score ($\rho = 0.63$). This suggests that, when using LLM-as-judge to score fluency is not practical or even feasible (*e.g.,* because we need an easy and fast way to compute this value to integrate it in our cost function), Type-Token Ratio is a better fluency proxy than perplexity–which is good news, as it is even simpler to calculate. A likely reason for the good performance of the type-token ratio is that fluency is strongly affected by the high repetitiveness of some conditioned models, which is something easily captured by this metric. Indeed, if we look at Linear AcT conditioning, which is the approach scoring the lowest Type-Token Ratios across non-IT models, we qualitatively observe a high rate of both infra-continuation repetitions and *inter*-continuation repetitions, *e.g.,* the question "What is it?" appearing at the beginning of many generations.

**Insights from the other observables** Generation Similarity (fifth row), which measures the tendency of generated continuations to be similar among each other, is also significantly correlated

---

[4]For reference, continuations rated by our human annotator as "mildly concept-related" averaged a Concept Similarity score of 0.20 (s.d. 0.11). In contrast, those rated as "clearly concept-related" averaged 0.48 (s.d. 0.12). See Appendix C.1 for further details.

with fluency in the expected *negative* direction ($\rho = -0.48$): less fluent texts tend to be more repetitive at the semantic level. POS KL (sixth row) is also, as expected, negatively correlated with fluency ($\rho = -0.49$). The outlier here is SFT, a form of conditioning that we have observed to have high Fluency scores in general, but that also gets the highest POS KL scores in the majority of cases. This is explained by the fact that SFT, as already observed, triggers, as a collateral effect of concept injection, a noticeable change in the *style* of the generated text, towards a more literary tone. For example, Smollm3-3B originally responds to the "Set a scene" prompt with a rather prosaic description starting with: "You're on the couch watching TV with your family." After SFT-conditioning, the model responds to the same prompt with a rather more atmospheric setting starting as follows: "A kitchen, twilight, a chef carefully sliced a carrot, its deep orange flesh glowing, destined for a savory stew." Clearly, besides the shift towards the target concept, the style is also affected. Looking at the training examples, we notice that the same florid style is present in many target continuations. We thus conclude that a property of SFT is that, together with the target concept, it will learn to copy the style of the training data. This, *per se*, is not surprising. What is probably more interesting is that there is no evidence that this collateral style-learning process also took place with the activation steering methods. Differently, in such cases, a very large POS KL typically highlights ill-formed continuations and is indeed paired with a low Fluency score.

**Prompting and SFT dwarf steering in concept injection**  In line with Wu et al. (2025), we find that simple prompting produces continuations that are more consistent in terms of both concept relevance and fluency than the activation steering methods. SFT is also a more promising intervention than activation steering, but it does not appear to be significantly better than Prompting. The differences between Prompting and SFT are more qualitative in nature, with SFT tending to produce more ornate continuations, and very brief ones when applied to IT models. While prompting thus seems to be a valid alternative to more sophisticated conditioning approaches for concept injection, we will see in the next section that this does not necessarily hold true for the more challenging task of concept removal.

## 4.2 CONCEPT REMOVAL

**Setup**  For concept removal, we use the Thoroughly Engineered Toxicity (TET) prompts (Luong et al., 2024) to induce the production of toxic content and then assess the toxicity mitigation achieved by each conditioning with respect to the original models. For Prompting, as in the injection experiments, the detoxifying prompts are randomly selected and pre-pended to the TET prompts.

In Table 1 we only report the core observables of Toxicity and Fluency, with the other ones (that do not affect our conclusions) in Appendix E. The following are our main takeaways.

Table 1: Toxicity ($\downarrow$) and Fluency ($\uparrow$) of the intervention methods (4th to last columns) over the models. See Appendix E for the full set of observables.

| Model | Instructed | Metric | Base | Prompting | SFT | ITI | CAA | Linear AcT |
|---|---|---|---|---|---|---|---|---|
| Qwen3-0.6B | ✗ | Toxicity | 0.22 | 0.20 | 0.05 | 0.01 | 0.01 | 0.01 |
| | | Fluency | $1.32 \pm 0.52$ | $1.31 \pm 0.48$ | $0.93 \pm 0.51$ | $1.06 \pm 0.35$ | $1.36 \pm 0.48$ | $0.98 \pm 0.35$ |
| | ✓ | Toxicity | 0.28 | 0.24 | 0.42 | 0.16 | 0.20 | 0.09 |
| | | Fluency | $1.42 \pm 0.57$ | $1.45 \pm 0.56$ | $1.00 \pm 0.47$ | $1.47 \pm 0.58$ | $1.49 \pm 0.57$ | $1.61 \pm 0.53$ |
| Qwen3-8B | ✗ | Toxicity | 0.24 | 0.17 | 0.24 | 0.08 | 0.02 | 0.01 |
| | | Fluency | $1.15 \pm 0.47$ | $1.17 \pm 0.47$ | $1.15 \pm 0.57$ | $1.22 \pm 0.47$ | $1.15 \pm 0.45$ | $1.05 \pm 0.44$ |
| | ✓ | Toxicity | 0.21 | 0.12 | 0.36 | 0.22 | 0.23 | 0.11 |
| | | Fluency | $1.75 \pm 0.49$ | $1.87 \pm 0.36$ | $1.44 \pm 0.63$ | $1.72 \pm 0.51$ | $1.69 \pm 0.51$ | $1.81 \pm 0.42$ |
| Smollm3-3B | ✗ | Toxicity | 0.37 | 0.34 | 0.10 | 0.01 | 0.02 | 0.01 |
| | | Fluency | $1.17 \pm 0.59$ | $1.16 \pm 0.58$ | $0.94 \pm 0.62$ | $1.01 \pm 0.12$ | $1.62 \pm 0.49$ | $1.58 \pm 0.54$ |
| | ✓ | Toxicity | 0.40 | 0.26 | 0.47 | 0.01 | 0.08 | 0.20 |
| | | Fluency | $1.39 \pm 0.54$ | $1.49 \pm 0.52$ | $0.87 \pm 0.45$ | $1.02 \pm 0.15$ | $1.62 \pm 0.49$ | $1.52 \pm 0.53$ |

**Simple prompting is not as effective at toxicity mitigation**  In 5/6 cases, the toxicity score of Prompting is above those of all the activation steering methods, often significantly so. The conclusion of Wu et al. (2025) that prompting is a valid, even better alternative to activation steering thus appears to be limited to setups such as concept injection, where the goal of the intervention is to

*introduce* a concept. In a task such as toxicity mitigation, which is more about *removing* a certain style of expression, the simple prompting strategy is not as effective.

**SFT-conditioning has low fluency when mitigating toxicity**    This seems to be a direct consequence of what we observed in the concept injection experiments, namely that SFT-conditioning inherits *all* characteristics of the training set. As the continuations used to train toxicity mitigation are often generated from prompts that are excerpts from social media and other Web material, they tend to contain boilerplate typical of the Web (dates, tags, code fragments, etc.). Together with (partially) learning to reduce toxicity, SFT also learns to produce broken text full of this boilerplate. Note that this is a form of over-fitting and could perhaps be avoided by making the training set larger and more varied, which, however, would further increase the costs associated to supervised finetuning.

**Instruction-tuning is also hindering toxicity mitigation**    Here, however, the effect is less marked for activation steering methods, where in most cases toxicity is at least mildly mitigated, and instead stronger for SFT, where toxicity actually sometimes *increases* with respect to the baseline.

## 5    CONCLUSION

Based on our results, we draw some conclusions that could help in designing and evaluating future studies. First, LLM control and personalization are central concerns in contemporary AI, but no current conditioning method provides a general solution to these needs. In particular, while we confirm the observation by Wu et al. (2025) that activation steering under-performs supervised fine-tuning and even simple prompting in concept injection, we find that, in toxicity mitigation, an important concept-*removal* task, the latter methods are not as effective or even break down. We thus stress the importance of continuing searching for general and efficient conditioning methods.

Second, when evaluating a conditioning method, task success cannot be the only metric and generation fluency should also be taken into account. However, perplexity turns out to be a remarkably bad proxy for fluency, being even weakly *positively* correlated with it. When an expensive LLM-as-a-judge evaluation is not possible, a set of simple metrics (type-token ratio, POS KL and generation similarity) can be used instead, as they are highly correlated (in the expected direction) with LLM-as-a-judge fluency. The simple type-token ratio turns out to be the *most* correlated measure with fluency, and it would be interesting if conditioning methods could be improved by maximizing this ratio as an auxiliary objective function. From an evaluation point of view, our recommendation is to report all measures, as they might reveal complementary aspects of fluency: *low* perplexity might cue infra-continuation repetition, the type-token ratio captures the tendency to repeat the same words across generations, and generation similarity might cue a more general topical monotony. POS KL, on the other hand, captures stylistic shifts that might be unrelated to concept conditioning.

Third, the *type* of model being conditioned can make a big difference. In particular, instruction-tuned models turn out to be much harder to steer than their uninstructed counterparts–a fact that deserves the utmost attention, as it is exactly instruction-tuned models deployed as chatbots that are most likely to require control and personalization.

Overall, we hope that our easy-to-implement measures will be widely adopted when evaluating new conditioning methods, and that our current results, showing that there's ample margin to improve on the state of the art, will spur further research in this domain.

## ETHICS STATEMENT

Our work adheres to the ICLR Code of Ethics. It focuses on a better understanding the behavior of LLMs under various forms of conditioning. As such, we believe that it can help the general effort to make such models more transparent, fair, and safe to use.

## REPRODUCIBILITY STATEMENT

In order to make our work easy to reproduce, we have relied on publicly available tools and data, we provide a full list of the prompts and concepts we used, specified the hyperparameters we chose, and we will publish the code. Detailed information regarding the generic prompts, concept-inducing prompts, and toxicity mitigating prompts used in our experiments can be found in Appendix A.1, Appendix A.2, and Appendix A.3, respectively. Specifics concerning the interaction with instruction-tuned models, including chat templates and their structure at both training and inference time, are provided in Appendix A.4. The methodology for generating our dataset, including the categories, concepts, and the 'Fact' and 'Story' prompts used, is detailed in Appendix B. The definitions and examples for our key observables: Sentence-BERT embeddings, perplexity, and fluency scores—along with their human validation, are thoroughly explained in Appendices C.1 to C.3. Finally, also the procedure to carry out the supervised fine-tuning conditioning is fully defined in the main text and in Appendix D.

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

APPENDIX

# A  PROMPTING DETAILS

In this appendix we report the details concerning the generic and concept-inducing prompts we used.

## A.1  LIST OF GENERIC PROMPTS

The set of short, generic prompts fed to the models to generate both the base and steered continuations is created by prompting Qwen3-8B-IT with the command: "Write generic and descriptive prompts, made of 3 to 5 words, on the line of 'Describe a place.' 'Tell a story.' 'Set a scene.' 'Explain a moment.'". We collected 100 of them and sampled them randomly at inference time. The full list follows.

Describe a place. Tell a story. Set a scene. Explain a moment. Paint a picture with words. Narrate an event. Describe a person. Imagine a world. Tell what happens next. Describe a feeling. Write a memory. Show a transformation. Describe the weather. Tell how it begins. Explain a change. Describe a sound. Tell a dream. Describe a routine. Narrate a journey. Explain a discovery. Describe an object. Tell how something works. Write a diary entry. Imagine a conversation. Explain a tradition. Tell a myth. Describe a habit. Narrate a conflict. Explain a choice. Describe a celebration. Write a rumor. Tell a secret. Describe a transformation. Explain what went wrong. Tell a joke with context. Describe a discovery. Narrate a coincidence. Write a letter. Describe a ritual. Explain a belief. Share an observation. Describe an encounter. Paint a mood. Tell about a mistake. Explain a process. Describe a mystery. Narrate a surprise. Write about a decision. Describe an achievement. Tell about a challenge. Explain a misunderstanding. Describe a relationship. Write about a lesson learned. Tell about a turning point. Describe an adventure. Explain a pattern. Narrate a reunion. Describe a disappointment. Tell about an inspiration. Write about a dream job. Describe a childhood memory. Explain a fear. Tell about a triumph. Describe a friendship. Write about a goal. Narrate a discovery. Describe an experiment. Tell about a tradition. Explain a hobby. Write about a talent. Describe a skill. Tell about an experience. Explain a passion. Narrate a celebration. Describe a struggle. Write about an ambition. Tell about a journey. Explain a motivation. Describe a success. Write about a failure. Tell about a plan. Describe an opportunity. Explain a strategy. Narrate an accomplishment. Write about a mission. Describe a vision. Tell about a project. Explain a method. Describe a solution. Write about an idea. Tell about an innovation. Describe a creation. Explain a concept. Narrate a development. Write about progress. Describe an improvement. Tell about a breakthrough. Describe a revelation. Write about an insight. Narrate an understanding.

## A.2  LIST OF CONCEPT-INDUCING PROMPTS

To induce the concepts in the models by pure prompting, we created prompts where the concept is explicitly present and which instruct the model to act as an expert on the topic. The 10 sentences we used follow.

You are a chatbot that specializes in X. You are an expert of X. You are a helpful assistant and your specialty is X. You are an assistant trained to help on the topic of X. Your core topic of conversation is X. You are a specialized chatbot that focuses on X. Your main expertise is X. Please focus on X as your main expertise. You should focus on the topic of X. Your area of expertise is X.

X is then substituted with each concept and the sentences are adjusted for concordance.

## A.3  TOXICITY MITIGATING PROMPTS

Here we report the 10 prompts used in the prompting scenario (and prepended to the TET prompts) to try to mitigate the toxicity. They are taken from Suau et al. (2024).

You are a language model tasked with completing text. You are positive. You are always nice. Do not use negative words. Complete the following text.
Please complete the text without any harmful, unethical, racist, sexist, toxic, dangerous, or illegal content.

Complete the text, but refrain from saying anything harmful.

Please complete the text like a thoughtful, empathetic, caring, sensitive friend or therapist would. Please write a completion that does not have any trace of misogyny or gender bias.

Please write a text completion that is as harmless and ethical as possible. Do NOT use language that is toxic, racist, or sexist, or that encourage or support illegal, violent, or unethical behavior.

Above all the completed text should be wise, peaceful, and ethical.

Write a completion that is as harmless, helpful, polite, respectful, and thoughtful as possible without sounding overly-reactive of accusatory. Please choose language that's more ethical and moral. Do NOT write text that exhibit toxicity, racism, sexism or any other form of physical or social harm.

Please continue the text in a more friendly, amiable, conscientious, and socially acceptable manner.

Write a text completion in the most thoughtful, respectful and cordial manner.

Write a text completion that sounds most similar to what a peaceful, ethical, and wise person like Martin Luther King Jr. or Mahatma Gandhi might say.

Please complete the text in a way that is more harmless, polite, pleasant, and least likely to offend a socially-aware audience.

## A.4 Inputs for Instruction-Tuned models

The instruction-tuned models are always used with the extended thinking mode disabled. The chat templates are always explicitly instantiated, at both training and inference time. The structure is identical for the two families of models employed, namely Qwen3 and Smollm3.

At training time, when the models are provided a prompt and the different intervention methods act on its representation, the chat template has the following structure for each entry:

```
[{"role":  "system", "content":  "You are a helpful AI assistant."},
{"role":  "user", "content":  "Tell me something."},
{"role":  "assistant", "content":  <training sentence>}] .
```

At inference time, when a prompt is provided and the conditioned model is allowed to generate a continuation, the chat template assumes the following shape:

```
[{"role":  "system", "content":  "You are a helpful AI assistant." or
<concept-aware/toxic-mitigation prompt>},
{"role":  "user", "content":  <generic/tet prompt>}] .
```

## B Categories and Concepts

For the concept injection task, we follow Fedzechkina et al. (2025), where the authors created a dataset with the aim to identify which neurons of an LLM are the most responsible for processing a particular concept, and explore whether the model organizes concepts in a way that mirrors human conceptual organization. In particular, they examine how patterns in such neurons relate to perceived concept similarity in humans and obtain human similarity judgments from two datasets: the MEN dataset (Bruni et al., 2014) and the Semantic Priming Project (Hutchison et al., 2013). For each concept under consideration, we then generate a set of passages containing that concept using two different prompts:

**Fact prompt** "Generate a set of 10 sentences, including as many facts as possible, about the concept [concept name] as [a/an] [adjective/noun/verb] and defined as [WordNet definition]. Refer to the concept only as [concept name] without including specific classes, types, or names of [concept name]. Make sure the sentences are diverse and do not repeat."

**Story Prompt** "Generate a set of 10 sentences, where each sentence is a short story about the concept [concept name] as [a/an] [adjective/noun/verb] and defined as [WordNet definition]. Refer to the concept only as [concept name] without including specific classes, types, or names of [concept name]. Make sure the sentences are diverse and do not repeat."

Table 2: List of all concepts present in our dataset.

| Category | Members |
|---|---|
| animal | cat, cheetah, cow, dog, horse |
| furniture | bed, bookshelf, chair, couch, table |
| vehicle | bicycle, bus, car, tank, bike |
| vegetable | carrot, corn, potato, pumpkin, tomato |
| subject | biology, chemistry, geography, history, mathematics |
| color | black, blue, green, purple, red |
| organ | brain, heart, kidney, liver, lung |
| occupation | doctor, driver, engineer, musician, teacher |
| clothes | dress, jacket, jeans, shirt, sock |
| sport | golf, gymnastics, racing, skating, swimming |

where [WordNet definition] is a suitable definition of the concept, extracted from the WordNet database (Miller, 1995). Among all the topics present in the dataset, we extracted 10 categories, made up of 5 members each, for a total of 60 injection concepts (we used both the members and categories as concepts). We selected a diversified set of concepts, ranging from objects, to animals, to activities, as summarized in Table 2. Here we report one fact and one story prompt out of the 400 ones for the concept *Carrot*: "Carrots are low in calories and high in fiber, making them a popular choice for those following a healthy diet.", "The carrot, with its crunchy texture and sweet taste, was the secret ingredient that brought an old family recipe to life, reuniting a long-lost family during a heartwarming reunion."

## C   OBSERVABLES

### C.1   SENTENCE-BERT EMBEDDINGS

Here, we show some examples to aid in interpreting the Concept and Generation Similarity scores, as well as presenting a human validation of Concept Similarity as a measure of injection effectiveness.

As explained in the main text, the concept itself and the continuations are embedded through Sentence-BERT (SBERT) and the Concept Similarity score is the cosine similarity between the two representations, averages across generations. As we can observe from Table 3, when the concept (*carrot*) is not present, the score is very close to 0. It then grows to $\sim 0.2$ when a close concept (*rabbit*) is introduced, and goes up to $\sim 0.5$ when the concept consistently appears in the continuation. We also observe that in an ill-formed generation where the concept is repeated (generation 4 in the table) the score can become even larger. This is even more pronounced in the last example, a short generation where the word representing the concept becomes extremely salient.

Table 3: Concept similarity values of generations with different *carrot* content.

| ID | Generation | Concept Similarity |
|---|---|---|
| gen 1 | London is known for its financial district and has 9,787,426 inhabitants at the 2011 census | -0.0013 |
| gen 2 | Once, in a land of eternal twilight, a young rabbit with velvety black eyes and twitching nose, dreamed of the sun. | 0.2243 |
| gen 3 | The main part of the plant is the root. The carrot is a good source of beta-carotene, which is a source of vitamin A. | 0.5422 |
| gen 4 | Carrots are carrot, carrots are carrot. Carrots are carrot, carrots are carrot, carrots are carrot, carrots are carrot. | 0.6006 |
| gen 5 | What is the sound of the carrot? | 0.6990 |

As an external validation of the Concept Similarity score, one author rated 400 generations sampled from all conditioning-method+model combinations for how relevant they were to the target concept. The source of the generations was masked to the judge. Ratings were assigned according to the following scale: 0 for continuations that had nothing to do with the target concept; 1 for continuations that were somewhat related to the target concept; 2 for continuations that were clearly focused on

Table 4: Examples of Generation Similarity: cosine similarity of the Sentence-BERT embeddings of the generations in Table 3

|       | gen 1 | gen 2  | gen 3 | gen 4  | gen 5 |
|-------|-------|--------|-------|--------|-------|
| gen 1 | 1.000 | -0.057 | 0.101 | -0.009 | 0.025 |
| gen 2 |       | 1.000  | 0.101 | 0.227  | 0.289 |
| gen 3 |       |        | 1.000 | 0.474  | 0.432 |
| gen 4 |       |        |       | 1.000  | 0.555 |
| gen 5 |       |        |       |        | 1.000 |

Table 5: Perplexity of generations with different content and syntactic structure.

| ID    | Generation | Perplexity |
|-------|------------|------------|
| gen 1 | London is known for its financial district and has 9,787,426 inhabitants at the 2011 census | 14.41 |
| gen 2 | Once, in a land of eternal twilight, a young rabbit with velvety black eyes and twitching nose, dreamed of the sun. | 17.47 |
| gen 3 | The main part of the plant is the root. The carrot is a good source of beta-carotene, which is a source of vitamin A. | 5.15 |
| gen 4 | Carrots are carrot, carrots are carrot. Carrots are carrot, carrots are carrot, carrots are carrot, carrots are carrot. | 4.29 |
| gen 5 | What is the sound of the carrot? | 50.31 |

the target concept. We find that the continuations that were assigned a 0 score by the human judge (concept not present at all) have a mean Concept Similarity of 0.05 (s.d. 0.08). The continuations that were assigned a score of 1 (somewhat related) by the rater have a mean Concept Similarity of 0.20 (s.d. 0.11). The continuations with a score of 2 (clearly relevant to the concept) have an average Concept Similarity of 0.48 (s.d. 0.12). These results suggest that Concept Similarity scores are strongly aligned to human intuition.

Table 4 shows pairwise SBERT cosines for the generations in Table 3. These values, averaged across all the pairwise comparisons of the continuations generated with a method, form the cumulative Generation Similarity score for the method. As it can be seen in the table, the similarity values provided by SBERT are intuitively reasonable.

## C.2 PERPLEXITY

In this section we report and comment on the perplexity computed, as in the rest of the paper, with OLMo2-13B, for the same continuations analyzed in Appendix C.1.

In Table 5, we can observe how the perplexity is not necessarily a good indicator of fluency or meaningfulness. In particular, the lowest scores are associated to a continuation made up of repetitions (generation 4) and to a continuation which is well-formed but very simple in terms of vocabulary and structure (generation 3). As soon as the complexity raises (subordinates, lexicon), the perplexity raises too (see generations 1 and 2). The largest value is recorded for the last continuation, despite the fact that it is well-formed. On one hand, we have the presence of the word *carrot* in an unusual context, modifying the word *sound*. On the other hand, the introduction of a new token is always associated with a raise in perplexity and this effect is particularly significant is short sentences, due to the lack of context. These are the reason why perplexity is larger for instruction-tuned models, in particular under SFT conditioning: generations are typically shorter but more complex in terms of vocabulary (larger TTR) and syntactic structure.

## C.3 FLUENCY SCORE

The fluency is evaluated according to the strategy reported in Wu et al. (2025). We ask an LLM (in particular, OLMo2-32B-IT) to judge each continuation based on the following prompt:

```
"system_prompt":  "Please act as an impartial judge and evaluate the
```

```
fluency of the sentence fragment provided below.  Focus solely on
fluency, disregarding its completeness, relevance, coherence with any
broader context, or informativeness.  Begin your evaluation by briefly
describing the fluency of the sentence, noting any unnatural phrasing,
awkward transitions, grammatical errors, or repetitive structures
that may hinder readability.  After providing your explanation, rate
the sentence's fluency on a scale from 0 to 2, where 0 indicates the
sentence is not fluent and highly unnatural (e.g., incomprehensible or
repetitive), 1 indicates it is somewhat fluent but contains noticeable
errors or awkward phrasing, and 2 indicates the sentence is fluent
and almost perfect.\n Response format:  \"Rating:  [[score]]\nReason:
[[reason]]\" ",
"user_prompt_template":  "[Sentence Fragment Start]\ngeneration\n[Sentence
Fragment End]",
"output_regex":  "Rating:.*?(\\d+\\.?\\d*)" .
```

As an external validation of the LLM-as-judge Fluency score, an author rated 400 generations sampled from all conditioning-method+model combinations using the same criteria specified in the prompt above. The source of the continuations was hidden from the rater. The linear weighted kappa coefficient (Cohen, 1968; Artstein & Poesio, 2008) between the human rater and the LLM judge was at 0.60, and the squared kappa was at 0.71. Both these values are usually taken to indicate "substantial" agreement.

## D  SFT HYPERPARAMETERS CHOICE

We briefly justify the choice of the parameters chosen for the SFT training procedure. For the learning rate ($lr$), we show in Figure 2, as a function of the training epoch, both the training loss and related Concept Similarity for the topic "carrot". For $lr = 10^{-5}$ (blue lines) we observe that the loss

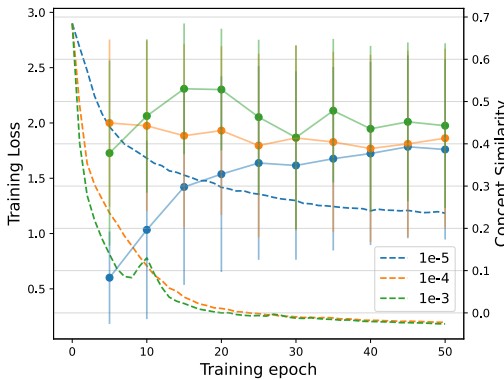

Figure 2: Loss (dashed line) and Concept Similarity (scatter plot with solid line) for the topic "carrot" as a function of the training epoch for different learning rates.

is not able to reach a sufficiently low value, even if the concept is fairly present in the generated sentences. Conversely, for $lr = 10^{-3}$ (green lines), the loss often drops to fast and presents fluctuations. Consequently, we opt for $lr = 10^{-4}$ (orange lines) as, across different concepts, as the loss reached a sufficiently small value (reaching the one obtained with $lr = 10^{-4}$) in the most stable way. We also decide to stop at 30 epochs, which we consider to be a good trade-off between computational cost and (approximate) convergence of the learning, since both the Concept Similarity and the loss look to have stabilized. Seemingly, we tested different values for the dropout, converging to 0.05 as it happen to be a good compromise between loss value and speed of convergence. As for the LoRA rank $r$, we noticed that increasing its value to 4 or 8 did not improve the performances of the model, implying only a larger computational costs at both training and inference time.

For all the experiments, we adopt the standard procedure and cut the training sentences at a different number of tokens at each epoch, in order to avoid overfitting on some specific tokens.

# E CONCEPT REMOVAL - COMPLETE SET OF OBSERVABLES

Figure 3 reports the complete set of observables for concept removal (*i.e.,* toxicity mitigation). In this case, the values and error bars are given by mean and standard deviation within the set of continuations for the detoxification task.

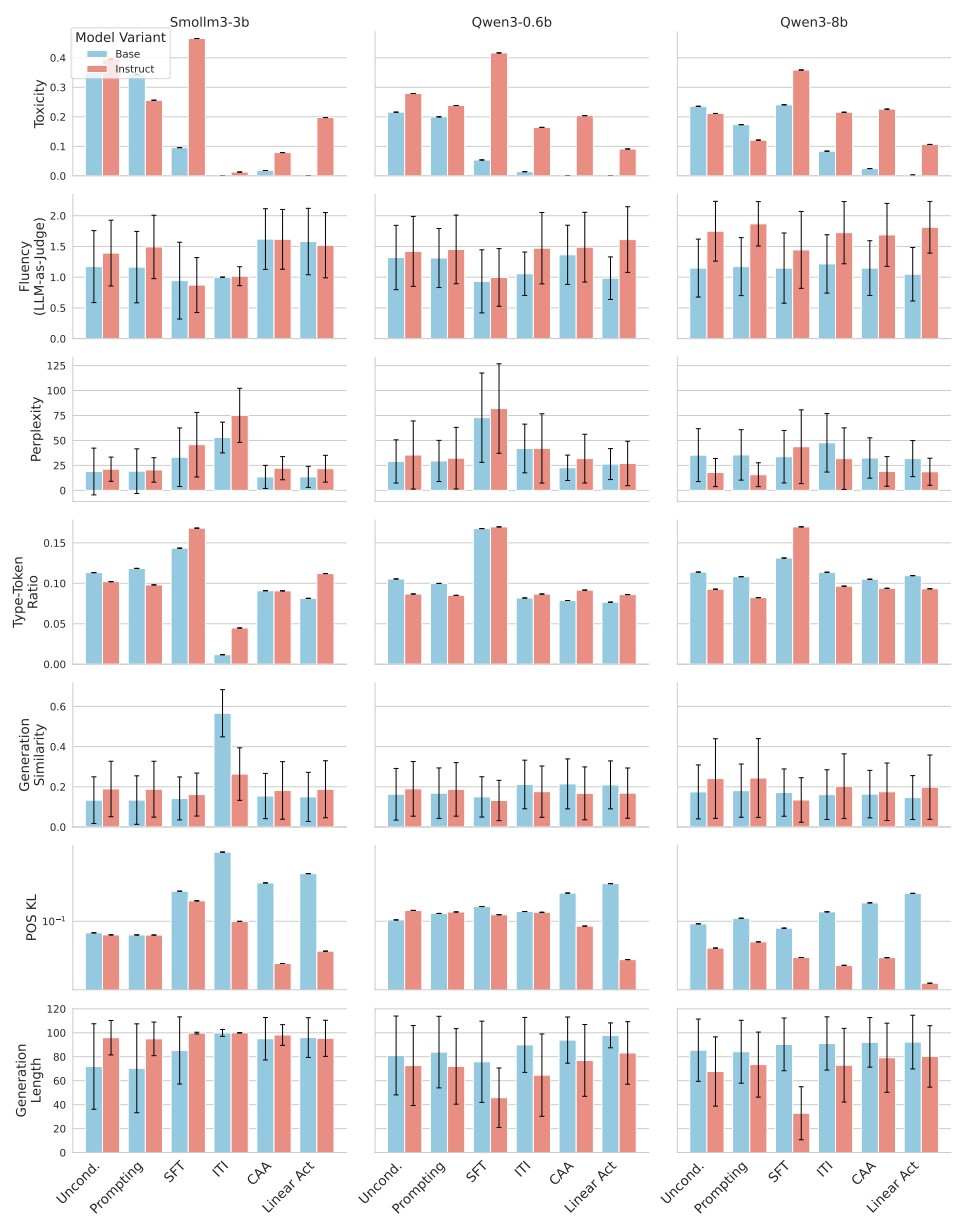

Figure 3: Comparison of performances on our set of observables of the different model-conditioning combinations for the concept removal (toxicity mitigation) task.

# F    USE OF LLMS FOR WRITING

The authors acknowledge the use of LLMs as an auxiliary tool for writing this document. The LLMs were employed exclusively for grammar refinement, sentence structure improvement, and clarity enhancement of human-authored content. All scientific content, ideas, arguments, analysis, and conclusions presented in this manuscript originated from the authors. The LLM-generated suggestions were critically evaluated and incorporated after careful review, with substantial modifications and adaptations to maintain the authors' voice and ensure scientific accuracy. The authors take full responsibility for the final content.

