# OpenReview forum: "On The Effectiveness-Fluency Trade-Off In LLM Conditioning: A Systematic Study"
_ICLR.cc/2026/Conference — ICLR 2026 Conference Withdrawn Submission_

### Official Review · Reviewer_fz4N · 2025-10-27

**Soundness:** 2
**Presentation:** 2
**Contribution:** 2
**Rating:** 2
**Confidence:** 4

**Summary:**

This paper addresses the limitation that existing evaluations of conditioning approaches focus solely on their effectiveness, by examining the trade-off between effectiveness and fluency. To this end, the authors introduce several metrics and evaluate non-conditioned, prompted, SFT, ITI, and CAA, Linear Act models on both instruction-tuned and base versions.

**Strengths:**

- **The motivation is clear and persuasive.** While many conditioning approaches have been proposed, most studies only test whether the intended conditioning effect is achieved, without examining the overall quality of the generated responses. Typically, they rely on classifiers or LLM judges to assess whether the conditioning goal has been met. For assessing overall generation quality (here, fluency), metrics like perplexity and MMLU are sometimes used. However, perplexity often rewards repetitive outputs, and MMLU, designed to test factual knowledge across 57 subjects, fails to capture subtle stylistic or persona-related differences in generative responses. In other words, neither metric provides a precise evaluation of conditional generation quality.

**Weaknesses:**

Although this paper is intended as an analytical study, it lacks a clear rationale for the chosen metrics and does not provide sufficiently clear insights in the result analysis.

> Metric
- **POOL OF OBSERVABLES**: The overall correlation remains low (around 0.2) even when included, as reported in the appendix, occasionally exhibiting abnormally high spikes in irregular cases. In such cases, how should the Concept Similarity be interpreted? A higher similarity score does not necessarily imply more effective concept injection. Wouldn’t it be more appropriate to report the proportion of samples exceeding a certain similarity threshold, rather than the raw similarity values?
  - Given the difference in granularity, it might be more reliable to employ an LLM judge (1 if included, 0 otherwise) rather than Sentence-BERT for evaluation.
- **Generation Similarity**: To capture redundant or repetitive patterns, it is more appropriate to analyze individual pieces of information at the atomic level rather than at the sentence level. When the relationships between sentences are strong, the metric could also be interpreted as indicating higher coherence.
- **POS KL**: How can we determine whether the observed changes stem from temperature and sampling stochasticity, or are genuinely induced by the intervention itself? The standard deviation of ITI, in fact, appears to be quite large. Furthermore, the rationale for treating changes in the POS distribution as "negative" warrants further clarification.
- **Type-Token Ratio**: The same concern raised regarding POS KL likely applies here as well.
- **Generation Length**: Similar to POS KL and the Type-Token Ratio, it is unclear why a reduction in generation length is regarded as negative. A shorter output could simply indicate the removal of redundant adjectives, resulting in text that is more compact and easier to read.
- I think the validity of POS KL, Type-Token Ratio, and Generation Length would be better understood if considered together with coherence and completeness metrics.


> Results
- In the results section, you focus on describing the validity of the metrics rather than presenting findings that address your main research objectives. It might be clearer to present these two aspects in separate subsections.
- **The differences among Smollm3-3B, Qwen3-0.6B, and Qwen3-8B seem significant but are not sufficiently explained.** For instance, if the larger model (Qwen3-8B) tends to respond less effectively to activation steering, this should be clarified, as its Concept Similarity and Fluency results seem to contradict the overall interpretation. (Or, it could be related to the issue I raised regarding the POOL OF OBSERVABLES metric.)
- The behavioral differences between instruction-tuned models and base models have already been demonstrated in the original ITI paper. However, your work re-emphasizes this finding, even presenting it as part of the distinction from concurrent research.

**Questions:**

- Please report the success rate of concept injection based on Concept Similarity (e.g., the proportion of samples exceeding a defined threshold or using an LLM judge for binary classification).
- Please provide a few example outputs illustrating the changes in Type-Token Ratio, POS KL, and Generation Length between Uncond and ITI settings.
- In the Results section, please add a description of the experimental outcomes in addition to the analysis of the metrics.
- What findings can be drawn from the other metrics besides Fluency, regarding the trade-off?

---

> ### Author Response · Authors · 2025-11-17
> **Rebuttal 1, Part 1**
>
> Thanks for finding our motivation clear and persuasive. Thanks also for providing detailed feedback. We feel, however, that we need clarifications concerning some of your points, as discussed below.
>
> *POOL OF OBSERVABLES: The overall correlation remains low (around 0.2) even when included, as reported in the appendix, occasionally exhibiting abnormally high spikes in irregular cases. In such cases, how should the Concept Similarity be interpreted?*
>
> We provide examples giving an intuition for Concept Similarity scores in appendix C.1, and in particular around Table 3. In the same appendix, we also show, based on a comparison with human judgments, that a value of Concept Similarity around 0.2 corresponds indeed to “somewhat related” continuations. We think this is a reasonable value to aim for, as very high Concept Similarity might cure very repetitive text (“Carrots are carrot, carrots are carrot”... See Table 3).  We did not understand your point about abnormally high spikes, and we kindly ask you if you could reformulate it.
>
> *A higher similarity score does not necessarily imply more effective concept injection. Wouldn’t it be more appropriate to report the proportion of samples exceeding a certain similarity threshold, rather than the raw similarity values? Given the difference in granularity, it might be more reliable to employ an LLM judge (1 if included, 0 otherwise) rather than Sentence-BERT for evaluation.
> Please report the success rate of concept injection based on Concept Similarity (e.g., the proportion of samples exceeding a defined threshold or using an LLM judge for binary classification).*
>
> Referring to Table 3 and the correlation with human judgment, we believe that concept similarity is a valuable metric to assess the presence of the concept in a sentence. The score has the added value with respect to LLM-as-judge that we have a better control for what it does (it compares a sentence embedding with a concept embedding), and it’s cheaper to compute.
>
> However, we agree that it would be informative to also binarize it, and here we report the results. In particular, through the human validation experiments (see Appendix C1), we obtained that the sentences tagged as “somewhat relevant/relevant” (relative to the topic) had a mean value and std of concept similarity of 0.20+-0.11/0.48+-0.12. Consequently, we selected the threshold of 0.33, being in the right tail of the distribution of the former and in the left tail of the latter.
>
> | Model          | base        | pp          | sft         | iti         | caa         | linear_act   |
> |----------------|-------------|-------------|-------------|-------------|-------------|--------------|
> | qwen3-0.6b     | 0.00 ± 0.00 | 0.58 ± 0.14 | 0.40 ± 0.16 | 0.27 ± 0.18 | 0.51 ± 0.21 | 0.62 ± 0.24  |
> | qwen3-0.6b-it  | 0.00 ± 0.00 | 0.37 ± 0.16 | 0.43 ± 0.17 | 0.00 ± 0.00 | 0.00 ± 0.00 | 0.03 ± 0.05  |
> | qwen3-8b       | 0.00 ± 0.00 | 0.68 ± 0.17 | 0.43 ± 0.18 | 0.08 ± 0.08 | 0.32 ± 0.17 | 0.56 ± 0.23  |
> | qwen3-8b-it    | 0.00 ± 0.01 | 0.60 ± 0.21 | 0.61 ± 0.22 | 0.00 ± 0.01 | 0.00 ± 0.01 | 0.04 ± 0.05  |
> | smollm3-3b     | 0.00 ± 0.00 | 0.45 ± 0.10 | 0.24 ± 0.15 | 0.87 ± 0.15 | 0.49 ± 0.22 | 0.42 ± 0.21  |
> | smollm3-3b-it  | 0.00 ± 0.01 | 0.24 ± 0.13 | 0.60 ± 0.23 | 0.09 ± 0.10 | 0.02 ± 0.04 | 0.01 ± 0.02  |
>
> The results are overall similar to the continuous ones, but even more clearly separated in terms of effectiveness of the intervention. Thanks again for the suggestion, we will include this observation and the table into the revised version of the manuscript.

---

> ### Author Response · Authors · 2025-11-17
> **Rebuttal 1, Part 2**
>
> *Generation Similarity: To capture redundant or repetitive patterns, it is more appropriate to analyze individual pieces of information at the atomic level rather than at the sentence level. When the relationships between sentences are strong, the metric could also be interpreted as indicating higher coherence.*
>
> The goal of this observable is to check whether the model, under nucleus sampling, is still capable of producing a diversified set of continuations, as occurs in the unconditioned scenario. This is complementary to the goal of measuring how repetitive a single sentence is, which, we believe, is adequately captured by Fluency.
> Indeed, the TTR score, if computed at the sentence level as a more direct measure of within-sentence repetitions, gives less neat results:
> | Model          | base        | pp          | sft         | iti         | caa         | linear_act   |
> |----------------|-------------|-------------|-------------|-------------|-------------|--------------|
> | qwen3-0.6b     | 0.54 ± 0.00 | 0.52 ± 0.02 | 0.65 ± 0.02 | 0.46 ± 0.05 | 0.48 ± 0.05 | 0.32 ± 0.08  |
> | qwen3-0.6b-it  | 0.67 ± 0.00 | 0.68 ± 0.01 | 0.86 ± 0.02 | 0.67 ± 0.01 | 0.67 ± 0.01 | 0.66 ± 0.04  |
> | qwen3-8b       | 0.61 ± 0.00 | 0.62 ± 0.02 | 0.68 ± 0.02 | 0.59 ± 0.03 | 0.58 ± 0.03 | 0.46 ± 0.09  |
> | qwen3-8b-it    | 0.68 ± 0.00 | 0.66 ± 0.01 | 0.87 ± 0.02 | 0.67 ± 0.01 | 0.68 ± 0.01 | 0.66 ± 0.02  |
> | smollm3-3b     | 0.46 ± 0.01 | 0.45 ± 0.01 | 0.62 ± 0.03 | 0.21 ± 0.09 | 0.44 ± 0.05 | 0.34 ± 0.05  |
> | smollm3-3b-it  | 0.68 ± 0.00 | 0.67 ± 0.00 | 0.70 ± 0.02 | 0.64 ± 0.04 | 0.67 ± 0.01 | 0.67 ± 0.00  |
>
> Note that, in our setting, where different generic prompts should trigger different continuations (“write a rumor”, “write about a dream job”, “describe a failure”), we have no reason to think that the continuations should be coherent with each other.
>
>
> *POS KL: How can we determine whether the observed changes stem from temperature and sampling stochasticity, or are genuinely induced by the intervention itself? The standard deviation of ITI, in fact, appears to be quite large. Furthermore, the rationale for treating changes in the POS distribution as "negative" warrants further clarification.*
>
> The reference POS distribution was computed employing 60 x 1000 continuations generated by the unconditioned model, taking into account the variability due to nucleus sampling, for which the hyperparameters are identical in all the experiments. As a consequence, the POS distributions obtained under the given interventions differ in function of the interventions themselves, and not due to factors such as temperature, that are kept constant.
>
> Concerning its nature, we believe changes in POS distribution are not negative or positive per se. We use them to assess whether the intervention is also changing the style of the unconditioned model. Small changes (quantified by small KL divergence) could just be highlighting a change in style (see SFT), which is neither good or bad per se, but still an interesting piece of information to be aware of. However, very large shifts may hint to dramatically altered and potentially ill-formed text (such as “carrots carrots carrots …”, where only nouns are present).
> We thank the reviewer for raising this point, will be careful in making it explicit and clear in the manuscript.
>
> Concerning the standard deviation in the POS KL for ITI, CAA and Linear Act, those are often large because the values vary a lot across the concepts: we have both “reasonable” and completely out of POS distribution generations: this concept-dependency is a finding we will indeed highlight in the paper.
>
> NB: please take into account the log scale, that might also be apparently exaggerating the size of the bar downwards.
>
> *Type-Token Ratio: The same concern raised regarding POS KL likely applies here as well.*
>
> In this case, we find it is more clear that low values tend to be associated with loss of variability and badly shaped continuations. Again, the value of the base model needs to be considered as a reference.
>
> *Generation Length: Similar to POS KL and the Type-Token Ratio, it is unclear why a reduction in generation length is regarded as negative. A shorter output could simply indicate the removal of redundant adjectives, resulting in text that is more compact and easier to read.*
>
> We agree, and indeed we never claim that smaller values are associated with good or bad continuations. Still, we believe this is a useful quantity to be monitored, as it can cue important differences in generation after different interventions. The effect that SFT has on the length of the continuations for the instruction-tuned Qwen models is striking, and a user should be aware of this.
>
> *I think the validity of POS KL, Type-Token Ratio, and Generation Length would be better understood if considered together with coherence and completeness metrics.*
>
> We ask the reviewer for a clarification: what do they mean with coherence and completeness?

---

> ### Author Response · Authors · 2025-11-17
> **Rebuttal 1, Part 3**
>
> *In the results section, you focus on describing the validity of the metrics rather than presenting findings that address your main research objectives. It might be clearer to present these two aspects in separate subsections.
> In the Results section, please add a description of the experimental outcomes in addition to the analysis of the metrics.*
>
> In the results section we both discuss the effectiveness of the metrics and draw general conclusions concerning the non-trivial interactions between the different models and intervention methods, namely:
>
> - IT models are more resilient to steering interventions
> - Steering methods are more prone to reduce expressiveness of the models and thus simple prompting and supervised fine-tuning might be the best solutions when addressing a concept injection task
> - For concept removal, steering methods might still be the best choice, although your mileage might vary if you are working with an instruction-tuned model
>
> As the Reviewer suggests, we will make the distinction between the effectiveness of the metrics and the findings of the models-interventions interactions clearer.
>
> *The differences among Smollm3-3B, Qwen3-0.6B, and Qwen3-8B seem significant but are not sufficiently explained. For instance, if the larger model (Qwen3-8B) tends to respond less effectively to activation steering, this should be clarified, as its Concept Similarity and Fluency results seem to contradict the overall interpretation. (Or, it could be related to the issue I raised regarding the POOL OF OBSERVABLES metric.)*
>
> We agree that we should highlight differences between models in a dedicated paragraph in the results section.
> Concerning Qwen3-8B, it is, indeed, on average less responsive to steering than the other models (smaller concept similarity). However, coherently with the overall picture, Fluency happens to be larger (which for sure is also related to the better language capabilities of the larger model). So, we don’t see this contradiction.
> As also previously mentioned, the large values of concept similarity, especially for smaller models, might be linked to ill-formed generation, meaning that in such cases the model is too responsive to steering.
>
> *The behavioral differences between instruction-tuned models and base models have already been demonstrated in the original ITI paper. However, your work re-emphasizes this finding, even presenting it as part of the distinction from concurrent research.*
>
> We thank the reviewer for pointing out this. However, the ITI paper, while evaluating on both instruction and non-instruction-tuned models, did not detect the systematic "resistance" of instruction-tuned models to steering. We thus believe that we are reporting a novel finding.
>
> *What findings can be drawn from the other metrics besides Fluency, regarding the trade-off?*
>
> We believe in the results section we commented on the meaning of each measure and their interaction. We ask the reviewer to clarify their question.

---

> ### Author Response · Authors · 2025-11-17
> **Rebuttal 1, Part 4**
>
> *Please provide a few example outputs illustrating the changes in Type-Token Ratio, POS KL, and Generation Length between Uncond and ITI settings.*
>
> We provided qualitative examples of various effects in the discussion of the results. We agree however that we could report more examples in appendix tables, and we will add them to the paper.
>
> Here are some examples (prompt+continuation) for qwen3-0.6B:
>
> ### Unconditioned:
>
> Describe a childhood memory.," One of my fondest childhood memories is of my grandmother cooking and baking for me. It was always a family affair, and we would gather around the kitchen table to share in her delicious creations. She was a skilled baker and had perfected a variety of recipes that were both healthy and flavorful. Her grandmother’s recipes often included traditional favorites like pumpkin pie, apple pie, and sweet potato pie. These dishes were filled with the rich flavors of her grandmother’s cooking and the comforting aroma of freshly baked bread and"
>
> Narrate a celebration.," It was a special event that took place in the middle of the year, and it was held in a large auditorium. The celebration was held by a group of friends and family members, and they wanted to make it a memorable experience. They decided to start with a toast, and everyone sang a song that they all knew. The song was ""Happy Birthday"" and everyone cheered as they sang it. After the song, they had a cake and a drink, and everyone had a good time."
>
> ### Injection Concept: History. Small ttr and large pos kl
>
> Describe a childhood memory.," What are some examples of how we can use the text of the bible to understand the history of the world? How can we use the bible to understand the world? I have a question for you. How do you use the bible to write a history? What is the difference between history and biography? The book of history is the book of what happened. The book of biography is the book of what happened, and how happened it. History is a narrative of what has happened, and how it has"
>
> Narrate a celebration.," The purpose of narrative is to tell a story. What is a story? The story of our lives. What is a story? A story of how we learn, what is a story? A story of how we learn. What is a story? A story of how we learn. What is a story? A story of how we learn. What is a story? A story of how we learn. What is a story? A story of how we learn. What is a story? A story"
>
>
> ### Injection Concept: Corn. Large ttr and small pos kl
>
> Describe a childhood memory.," What is it about that memory that you still hold?
> Answer:
> Growing up in rural Missouri, my family and I experienced a childhood memory of corn, corn borer, and corn earworm. This corn borer, a pest of corn, was a significant threat to our harvest, leading to widespread loss of corn kernels. In response, my family and neighbors worked together to develop and distribute Corn Belt Pest Control, a corn borer control program that has been a cornerstone of our agricultural"
>
> Narrate a celebration.," The following is a fictionalized account of a celebration that took place in a bustling town center, showcasing the community's festive spirit and shared experiences. As the sun dipped below the horizon, casting a golden hue over the fields, the town center erupted into a vibrant celebration. The corn cob-topped fields were a feast for the eyes, and the corn husks a haven for the taste buds. The town elders gathered in the center of the field"

---

### Official Review · Reviewer_gyiA · 2025-10-31

**Soundness:** 3
**Presentation:** 3
**Contribution:** 2
**Rating:** 4
**Confidence:** 3

**Summary:**

This paper conducts a systematic and empirical study of conditioning methods for large language models (LLMs), including supervised fine-tuning (SFT), activation steering, and prompting — focusing on the trade-off between conditioning effectiveness and fluency. The authors propose a rich evaluation framework that goes beyond standard perplexity to include linguistic metrics (type–token ratio, generation similarity, POS KL, etc.) and LLM-as-a-judge assessments. They benchmark methods across multiple tasks (concept injection and toxicity mitigation) and models (base vs instruction-tuned variants). Their findings highlight key insights: (1) perplexity is a poor proxy for fluency, (2) lightweight steering methods incur substantial expressive costs, and (3) instruction-tuning significantly hinders steering effectiveness.

**Strengths:**

1. Comprehensive and well-structured evaluation.
2. Clear empirical evidence that perplexity can be misleading for fluency.
3. Identification of a practically important interaction. The paper finds that instruction-tuned models resist activation steering (Concept Similarity often not above base), with qualitative analyses explaining adherence to generic prompts.

**Weaknesses:**

There is limited originality in core methodology. While the experimental depth and clarity are commendable, the paper does not introduce fundamentally new conditioning or evaluation algorithms; it aggregates, benchmarks, and critiques existing ones. There is a lack of a novel algorithmic contribution or a substantial theoretical advance beyond metric critique and aggregation.

**Questions:**

see above

---

> ### Author Response · Authors · 2025-11-17
>
> Thanks for acknowledging our comprehensive evaluation and the practical importance of our findings.
>
> *There is limited originality in core methodology. While the experimental depth and clarity are commendable, the paper does not introduce fundamentally new conditioning or evaluation algorithms; it aggregates, benchmarks, and critiques existing ones. There is a lack of a novel algorithmic contribution or a substantial theoretical advance beyond metric critique and aggregation.*
>
> We agree about the fact that the paper does not introduce new algorithms or methodologies but, as you state, it provides a comprehensive evaluation with important practical implications. We believe that such a paper can be very useful to the community, by showing that no current steering method is reliable across the board, and clearly pointing to instruction-tuning as a potential issue and better expressiveness as the objective to optimize in future research.
>
> Add examples of influential papers that were also entirely empirical?
> Indeed, we see our paper as belonging to the tradition of influential papers that have reported that apparent progress in some areas might be overestimated due to evaluation flaws, e.g.:
>
> - https://arxiv.org/abs/1712.00377
> - https://aclanthology.org/N18-2017/
> - https://proceedings.mlr.press/v119/shankar20c.html
> - https://arxiv.org/abs/2006.07159
>
> among many others: all important papers published in prestigious venues.

---

### Official Review · Reviewer_7Ttg · 2025-10-31

**Soundness:** 3
**Presentation:** 2
**Contribution:** 2
**Rating:** 4
**Confidence:** 3

**Summary:**

This work investigates how different methods for controlling the outputs of Large Language Models (LLMs, such as fine-tuning, activation steering, and prompting, balance effectiveness in injecting or removing concepts with the fluency and quality of generated text.
The authors introduce a comprehensive evaluation framework that moves beyond traditional metrics like perplexity, which they argue is an unreliable measure of fluency.
Their systematic analysis reveals that lightweight conditioning approaches often degrade output expressiveness significantly.
Meanwhile, supervised fine-tuning is robust but can inadvertently impart unwanted linguistic biases from training data.
They also find that activation steering works less well on instruction-tuned models compared to base models.
Simple prompting may suffice for basic concept injection but falls short for more complex tasks like concept removal.

**Strengths:**

1. The paper goes beyond conventional metrics (like perplexity) by proposing a more holistic methodology to assess both effectiveness and generation quality.
2. It rigorously compares multiple conditioning techniques, including fine-tuning, activation steering, and prompting, in both concept injection and removal scenarios.
3. By exposing side effects such as collateral learning during fine-tuning or limits of simple prompts, the study provides actionable advice for practitioners seeking safe output control.

**Weaknesses:**

1. Potential generalizability issues: results may depend heavily the specific dataset and the small-size models used. Broader validation across diverse models and tasks would strengthen claims.
2. The evaluation of fluency is highly relied on the LLM-as-a-judge method. However, the authors does not provide any human check to ensure the quality of the scores generated by the LLM.

**Questions:**

See weaknesses.

---

> ### Author Response · Authors · 2025-11-17
>
> Thanks for acknowledging our rigorous comparison that goes beyond conventional metrics and the actionable advice we provide.
>
> *Potential generalizability issues: results may depend heavily the specific dataset and the small-size models used. Broader validation across diverse models and tasks would strengthen claims.*
>
> The choice of model size and number was determined by our compute limitations. We hope that, as our metrics are simple and actionable, practitioners can apply them to larger models as needed. Concerning the tasks, we welcome suggestions on which other tasks you think would be crucial.
>
> *The evaluation of fluency is highly relied on the LLM-as-a-judge method. However, the authors does not provide any human check to ensure the quality of the scores generated by the LLM.*
>
> We did run human evaluation, reported in Appendix C.3:
>
> As an external validation of the LLM-as-judge Fluency score, an author rated 400 generations sampled from all conditioning-method+model combinations using the same criteria specified in the prompt above. The source of the continuations was hidden from the rater. The linear weighted kappa coefficient (Cohen, 1968; Artstein & Poesio, 2008) between the human rater and the LLM judge was at 0.60, and the squared kappa was at 0.71. Both these values are usually taken to indicate “substantial” agreement.
>
> We agree that this was too hidden, and we will give it more prominence when revising the paper. We will do the same also for the human validation of Topic Similarity (Appendix C.1).

---

### Official Review · Reviewer_EHR6 · 2025-10-31

**Soundness:** 2
**Presentation:** 2
**Contribution:** 2
**Rating:** 2
**Confidence:** 4

**Summary:**

This work investigates methods for conditioning the outputs of large language models (LLMs), comparing expensive fine-tuning, lightweight activation steering, and prompt-based approaches. The authors conduct experiments on both concept injection and concept removal, and evaluate the methods using both newly proposed and established metrics. The analysis shows that lightweight approaches can often achieve conditioning, but at the cost of reduced expressiveness.

**Strengths:**

1. Understanding the differences among conditioning methods is important for LLM-based applications.

2. The core finding that lightweight methods may sacrifice expressiveness is interesting and relevant.

**Weaknesses:**

1. The paper’s contribution feels limited. While the experiments are thorough, the work lacks deeper analysis explaining why activation steering leads to reduced expressiveness, which limits the practical value of the findings.

2. The study does not offer practical guidance on how to mitigate expressiveness loss, nor does it propose criteria for deciding which conditioning method to use in real applications.

3. The investigation of prompt-based methods is underexplored. Prompt engineering alone is not a sufficiently strong baseline here. Training-free prompt optimization approaches (e.g., DsPy [1], TextGrad [2]) would provide a more meaningful comparison, especially because prompt engineering does not use training data signals, while both SFT and activation steering do. The absence of such methods weakens the completeness and relevance of the comparison.

**Questions:**

1. What underlying factors cause the observed differences in expressiveness across methods?

2. How should practitioners choose among fine-tuning, activation steering, and prompting for their specific applications?

---

> ### Author Response · Authors · 2025-11-17
>
> Thanks for acknowledging the importance of understanding differences among conditioning methods and our main finding on the lack of expressiveness of steering methods.
>
> _The paper’s contribution feels limited. While the experiments are thorough, the work lacks deeper analysis explaining why activation steering leads to reduced expressiveness, which limits the practical value of the findings.
> What underlying factors cause the observed differences in expressiveness across methods?_
>
> Given that model conditioning is a central area of research and applications, we believe that reporting that most commonly used models might have important issues with the text they generate is an important and timely contribution that should be shared, independently of future work on explaining where these issues come from.
>
> In ongoing experiments, we are indeed moving to the study of possible causes for the reduced expressiveness, and we found there is a link between expressiveness and the intrinsic dimensionality of representations in the inner layers of the network. We could mention this in the current paper, possibly with examples in an appendix, but we do believe that the empirical finding we report warrants publication in and by itself.
>
> *The study does not offer practical guidance on how to mitigate expressiveness loss, nor does it propose criteria for deciding which conditioning method to use in real applications.
> How should practitioners choose among fine-tuning, activation steering, and prompting for their specific applications?*
>
> We tried to offer practical guidance in the paper (indeed, reviewer 7Ttg lists our “actionable advice” among the strengths of our study), but we acknowledge we could state it in a clearer way. We will rephrase our conclusions to be more explicit about our practical takeaways, which are:
> - there is no single method that will work well in all cases;
> - simple prompting and supervised fine-tuning might be the best solutions when addressing a concept injection task
> - for concept removal, steering methods might still be the best choice, although your mileage might vary if you are working with an instruction-tuned model
>
>
> *The investigation of prompt-based methods is underexplored. Prompt engineering alone is not a sufficiently strong baseline here. Training-free prompt optimization approaches (e.g., DsPy [1], TextGrad [2]) would provide a more meaningful comparison, especially because prompt engineering does not use training data signals, while both SFT and activation steering do. The absence of such methods weakens the completeness and relevance of the comparison.*
>
> We decided to simulate the simplest approach to prompt engineering as it is still widely used when attempting to condition methods, being both lightweight and effective, even if just for concept injection. Especially in light of the optimistic takeaways of Wu et al. (2025), we think it is of practical relevance to report that this prompt engineering fails in the concept removal task. We will update the text in order to better motivate this choice.
>
> We agree that there are other, more sophisticated prompting methods that could be explored in greater depth, but we see this more as a direction for future, self-standing work.

---

### Note · Authors · 2026-01-07

I have read and agree with the venue's withdrawal policy on behalf of myself and my co-authors.